# Recent Developments in Carbon Quantum Dots: Properties, Fabrication Techniques, and Bio-Applications

Rehan M. El-Shabasy [1], Mohamed Farouk Elsadek [2,3,*], Badreldin Mohamed Ahmed [2], Mohamed Fawzy Farahat [2], Khaled N. Mosleh [4] and Mohamed M. Taher [5]

[1] Department of Chemistry, Faculty of Science, Menoufia University, Shebin El-Kom 32512, Egypt; elshabasy1010@gmail.com

[2] Department of Community Health Sciences, College of Applied Medical Sciences, King Saud University, Riyadh 11433, Saudi Arabia; bmohamed@ksu.edu.sa (B.M.A.); mffarahat@ksu.edu.sa (M.F.F.)

[3] Nutrition and Food Science Department, Faculty of Home Economics, Helwan University, Helwan 11795, Egypt

[4] Vitane Implant International, 67100 Strasbourg, France; k.nafea@gmail.com

[5] Department of Chemistry, Faculty of Science, Cairo University, Cairo 12613, Egypt; m.taher923@gmail.com

* Correspondence: mfbadr@ksu.edu.sa

**Abstract:** Carbon dots have gained tremendous interest attributable to their unique features. Two approaches are involved in the fabrication of quantum dots (Top-down and Bottom-up). Most of the synthesis methods are usually multistep, required harsh conditions, and costly carbon sources that may have a toxic effect, therefore green synthesis is more preferable. Herein, the current review presents the green synthesis of carbon quantum dots (CQDs) and graphene quantum dots (GQDs) that having a wide range of potential applications in bio-sensing, cellular imaging, and drug delivery. However, some drawbacks and limitations are still unclear. Other biomedical and biotechnological applications are also highlighted.

**Keywords:** carbon dots; optical properties; green synthesis; biomedical applications; bioimaging

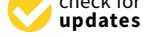



## 1. Introduction

In recent years, carbonaceous and carbon-based nanomaterials have gained great attention owing to their relevant properties [1–4]. In particular, these substances are characterized by high biocompatibility, less toxicity, significant thermal and mechanical features, and can functionalize easily [5–8]. Fluorescent carbons are commonly known as carbon dots because of their unique properties that revealed strong fluorescence [9]. In addition, carbon dots are distinguished by high stability, reducing toxic activity, water solubility, and derivatization availability. All of these unique features support their applications in several disciplines as shown in Figure 1 [10–14]. Carbon dots are relatively new and considered one of the most promising nanomaterials ever recognized to humanity, mainly composed of the heteroatoms (functional groups) attached with carbonized core [15]. Carbon dots including different types of nanomaterials such as polymer dots, carbon nanodots, and graphene quantum dots (GQDs). It defined as nanoparticles with small size (<10 nm) that consist of $sp^2$ hybrid conjugated of carbon core-shell between carbon (core) and organic functional groups (shell) such as N–H,–OH,–C = O, COOH, C−O, and C–N or polymer aggregates [16]. Several studies have reported that different techniques and carbon sources are employed in the fabrication of carbon dots with different structures [17].

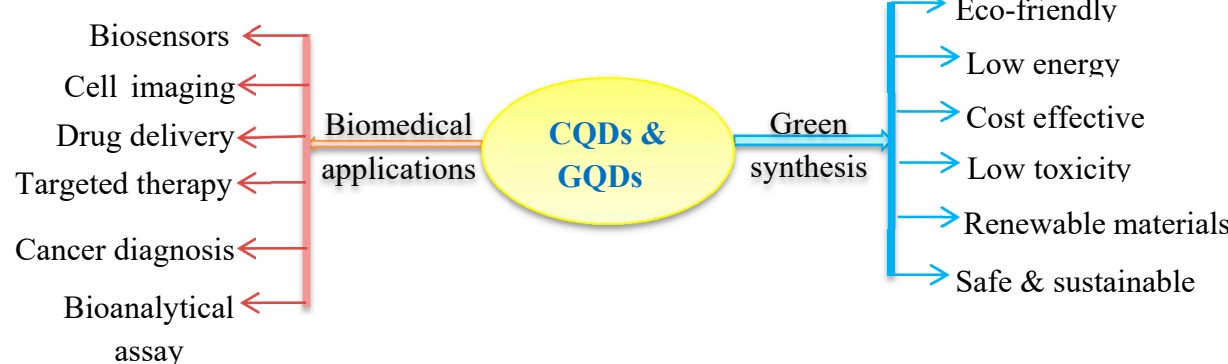

**Figure 1.** Chart of green synthetic methods of quantum dots (carbon quantum dots (CQDs) and graphene quantum dots (GQDs)) and their biomedical applications.

Typically, two techniques are commonly used in the formation of carbon dots; top-down and bottom-up as described in Figure 2 [18,19]. Usually in the first process "top-down" carbon dots are fabricated by chemical and physical cutting approaches; laser ablation/passivation [20], chemical oxidation, and electrochemical synthesis [21]. In the second method, "bottom-up" carbon dots are converted from appropriate molecular precursors with specific conditions represented by combustion, hydrothermal and thermal [22], and ultrasonic irradiation [23] in which the conditions required fewer amount of carbon sources. It is noteworthy to mention that bottom-up strategy is more preferable to top-down because some limitations are related to this technique including the high cost of the required materials, long time, and harsh conditions [24]. Further, the fabrication of carbon dots via top-down approaches usually needs a separate step for functionalization and passivation of the surface but the second method" bottom-up" does not require that [15]. In addition, different more approaches have been reported such as plasma treatment [25], cage-opening of fullerenes [26], and solution chemistry approaches [27]. The formed carbon dots, nevertheless, of its production methods, have different sizes that required complicated separation technique to get a mono-dispersed carbon dot. There are several post-synthesis separation processes such as chromatography [28], dialysis [29], and gel electrophoresis [30]. On the other hand, the characterized composition of carbon dots gave it significant points for several applications like bioimaging, label-free detection, photocatalysis, and sensing. The current review discussed synthetic approaches for the fabrication of carbon dots and the most significant properties. The biotechnological and biomedical applications are also highlighted.

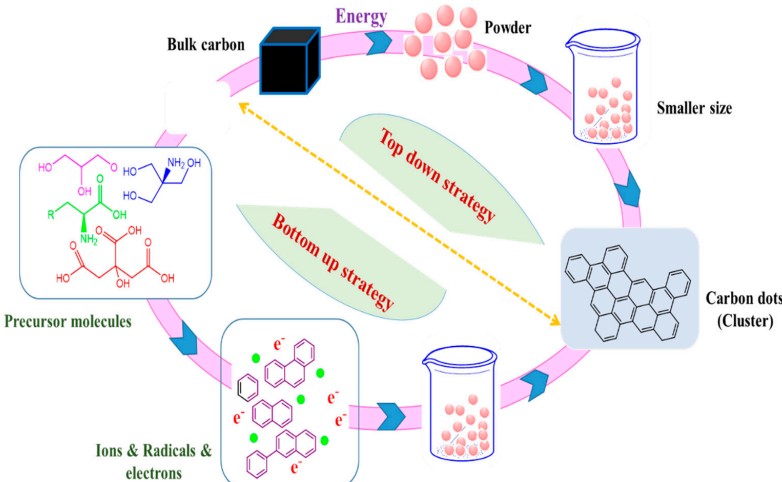

**Figure 2.** Generation of carbon dots by well-known Top-down and bottom-up approaches.

## 2. Unique Features of Quantum Dots (CQDs and GQDs)

Carbon dots are considered one of the recently discovered materials having promising and unique properties [31]. The chemical composition of carbon dots containing several function groups on the surface such as amino groups, oxygen, and polymer chains is highly supported by their remarkable features. These functionals have a significant effect on photoluminescence activity and also enhanced the energy gap and energy level of the surface [32]. Such substances have gained great attention because of their significant tunable optical properties, less toxic, simplicity, and low cost, which support them as perfect candidates for use in optical sensors [33]. The aptness of emission of light through carbon dots near the Infra-red area is of particular prominence because the light in this region has deeper tissue penetration proficiency and biological systems are transparent to these wavelengths [34]. Typically, CQDs and GQDs exhibit effectiveness in the short-wavelength area for photon-harvesting that caused by $\pi–\pi^*$ transition of C = C bonds and n-$\pi^*$ transitions of the groups; C–N, C = O, and C−S for example. Significant optical absorption was demonstrated in the ultraviolet region expanded to the visible range. The region between 230 and 270 nm appeared absorption owing to $\pi–\pi^*$ transition related to C = C bonds, while the peak shoulder in the range of 300–390 nm is attributed to n–$\pi^*$ transition of C = O bonds [35]. The absorbance can be modified by different types of surface passivation and functionalization methods [36]. For example, multimode emissive carbon dots with high fluorescent were prepared using D-cysteine and L-cysteine. Two absorption bands appeared at the same time related to L-carbon dots at 243 and 300 nm with the low band at 400 nm. The absorbance was displayed due to $\pi–\pi^*$ transition of the aromatic $sp^2$ domains (243 nm) and n-$\pi^*$ transition of C = O, C–N, C–S (300 nm). However, D-cysteine was not showed any band above 240 nm [37]. The results reported that several function groups (e.g., $NH_2$ and COOH) were found on the surface of L-carbon dots and hence the band gap increased due to the surface interfacial excitation. In addition, Lin et al. have recently investigated the synthesis of other carbon dots from poly (vinyl alcohol) and phenylenediamine. The formed composite exposed two different bands at 247 and 355 nm, matching to $\pi–\pi^*$ transition of C = C bonds and n-$\pi^*$ transition of C–N, C = N, respectively [38]. Commonly, CQDs have been evaluated successfully in surface passivation as they have the ability for improving brightness because of long wavelengths and decreasing quantum yield. On the opposite, the quantum yield of graphene dots was more than carbon quantum dots because their structures appeared as layers and crystalline phases [32]. The color of the fabricated carbon dots was changed between red, green, and blue. It was not recommended for multi-color imaging, due to the differences in chemical composition, size, and increasing heterogeneity of carbon dots. Most of these particles appeared wide emission spectra originating from difficulty controlling the synthesis processes. Interestingly, carbon dots have several attractive optical properties, but photoluminescence is the most significant one, including phosphorescence and fluorescence. The property of electrochemiluminescence plays an important role in surface passivation, whereas CQDs that passivated have a strong fluorescence and weak electrochemiluminescence [39]. For example, methyl parathion sensors were fabricated by the hydrothermal reaction between tyrosine methyl ester and carbon dots with citric acid employed as a resource of carbon. These types of sensors revealed high and stable photoluminescence and the yield of quantum was approximately 3.8%. This could be successfully developed to determine organophosphorus compound [40].

In addition, most studies revealed that carbon dots have excitation-dependent fluorescence features, although, the excitation-independent emission in S, N-co-doped carbon dots have been investigated [41]. For instance, excitation-independent carbon dots with tunable fluorescent colors have been synthesized through a well-controlled wet oxidative process whereas the results displayed that the photoluminescent properties of carbon dots were principally detected by surface oxidation degree and their molecular weight [42]. The fluorescent carbon dots having fluorescence wavelength can be tuned across the visible spectrum with varying the passivation or functionalization substances, the molar mass

ratio of the precursors, and the different synthetic factors. The emission of CQDs can be also influenced by an assortment of adaptable solvents. Subsequently, the performance of excitation dependent/independent photoluminescence is mostly originating from the surface states of carbon dots [43]. It is worth mentioning that the emission mechanism of carbon dots is still unclear. Currently, some expected theoretical explanations may be acceptable including surface state electron-hole radiation rearrangement, quantum size effect, and molecular state luminescence emission mechanism [44]. Consequently, the preparation of monochromatic fluorescent carbon dots and the study of the fluorescence mechanism is an imperative research area for developing the applicability of carbon dots.

On the other hand, biocompatibility is one of the most important features that showed a considerable influence on the application of carbon dots particularly in bio-imaging and cellular imaging [33]. GQDs having an excess of oxygen groups which showed high biocompatibility, low toxicity and enhanced for use in radiotherapy [45,46]. The cytotoxic effect of GQDs was caused by reactive oxygen species generated from the function groups. For example, the in vivo studies of GQDs exhibit low toxicity, no accumulation in the basic organs, and the kidney can dispose of it quickly. By investigation, it has appeared that the mice were not affected by injection with GQDs whereas the graphene oxide showed toxic activity until its death. This happens because graphene oxide can aggregate in the organs.

## 3. Fundamental Approaches of Carbon Dots Fabrication (Green Synthesis)

Green synthesis is Avery important topic matching with sustainability in our daily life [47–51]. Relevant studies have reported that small organic precursors can be polymerized and carbonized for the synthesis of carbon dots such as ammonium citrate [52], ethylene glycol [53], citric acid [54], phenylenediamine [55], graphite [56], and carbon nanotube [57]. To make them potential fluorescent materials with unique surface functionalities, two main approaches are widely investigated for the generation of ultra-small fluorescent carbon dots. Among these synthetic approaches, the colloidal synthetic methodology has received significant interest due to the generation of large quantities with a tightly controlled size of carbon dots [58]. For example, GQDs were fabricated from small aromatic molecules by stepwise solution chemistry and characterized by significant size uniformity and well-defined structures as presented in Figure 3 [59]. The structure controlling could be enhanced through the covalently bonding between 2',4',6'-trialkyl-substituted phenyl moieties (at the 1'-position) to the edges of graphene. The peripheral phenyl groups twisted from the plane of the graphene due to the crowding on the edges, and then the alkyl chains forming a three-dimensional cage around it (Figure 3). This action caused increasing distance between the conjugated systems in all three dimensions and consequently critically decreases the intermolecular $\pi$–$\pi^*$ attraction. The well-defined colloidal quantum dots have some unique characteristics that make them excellent model systems for studying fundamental processes in complex carbon materials.

Green chemistry is one of the fundamental branches of chemistry that provide golden solves for most problems with significant properties. Green chemistry has several advantages including, safe, environmentally friendly, no need for hazardous materials, and can occur under normal conditions [60]. Green chemical procedures have been engaged in the formation of carbon dots from several natural sources counting chicken eggs, animals [61], different plant species including fruits and vegetables [62], and waste materials like waste paper and frying oil [63]. There is an exponential increase in the number of research articles with both carbon quantum dots and green synthesis content. The fabrication processes can be achieved by different types of methods including hydrothermal/solvothermal, microwave-assisted polymerization, pyrolysis, and carbonization [64]. These approaches are widely used in the synthesis of carbon dots and having several advantages as displayed in Figure 4. In Table 1, conventional CQDs production methods, and maximum emission wavelength, quantum yield and reported CQDs dimensions of CQDs produced by these methods were compared.

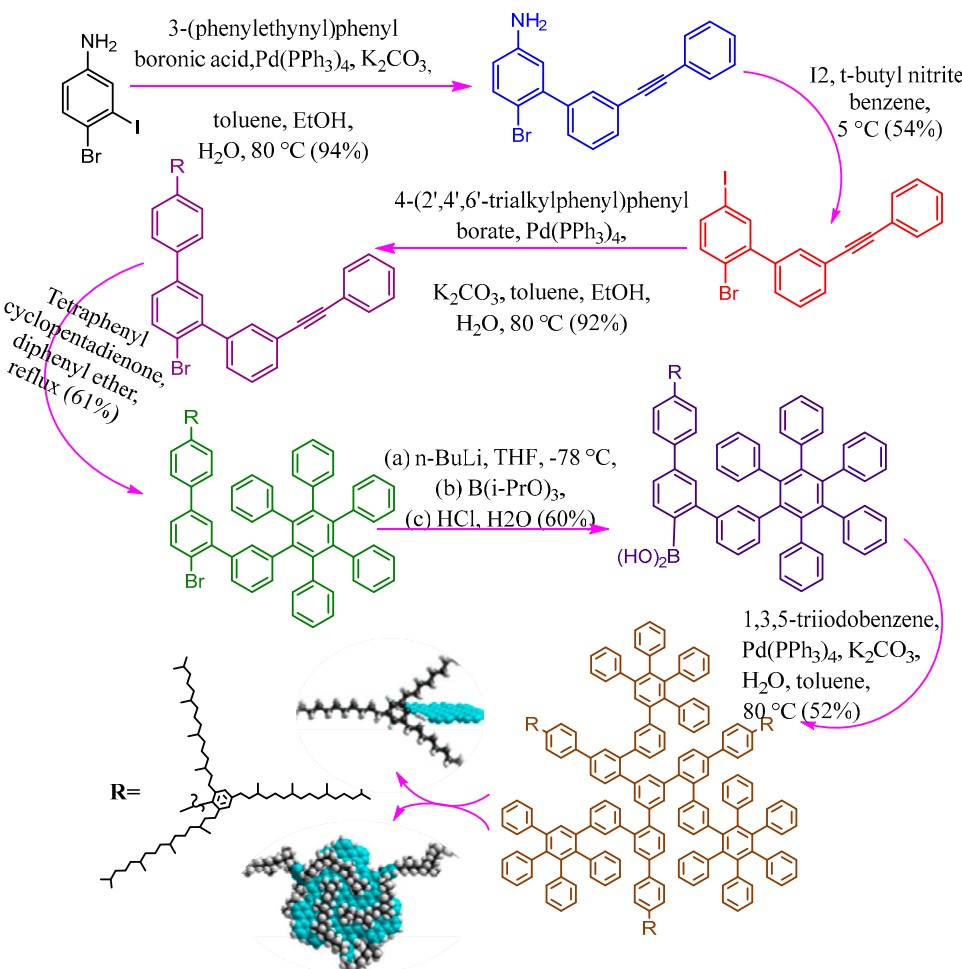

**Figure 3.** The synthetic procedures of colloidal GQDs with well-defined structures.

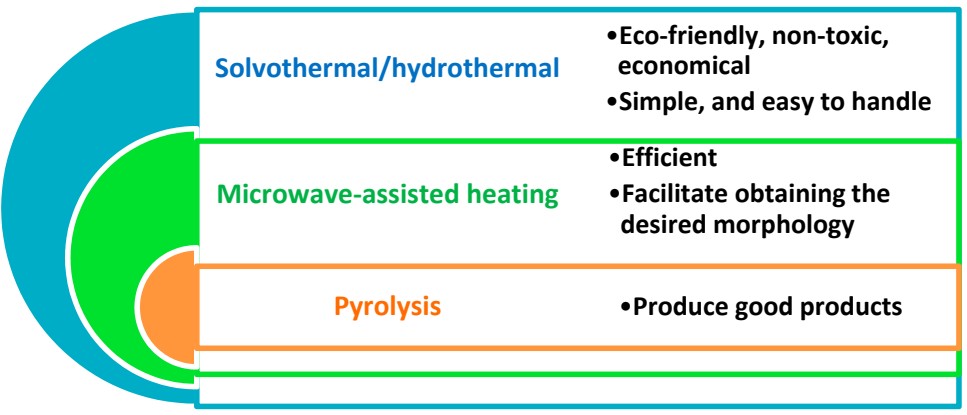

**Figure 4.** Advantages of different current routes used for the preparation of carbon dots.

**Table 1.** Recent green quantum dots produced by the bottom-up approach (Hydrothermal, microwave-assisted, and pyrolysis) and their applications.

| Synthetic Approach | Source | Quantum Yield (%) | Size Range (nm) | $\lambda_{em}$ Max | Application | Ref. |
|---|---|---|---|---|---|---|
| Hydrothermal | Banana peel waste | 5 | 4–6 | 355–429 | Bio imaging | [65] |
| Hydrothermal | Cambuci juice (*Campomanesia phaea*) | 21.3 | 3.7 | 270, 283 | Sensing of $Zn^{2+}$ | [66] |
| Hydrothermal | Biomass waste | 4.3–8.2 | 1.3 and 4.9 | 445, 435, 43, 435 | Detection of $Fe^{3+}$ | [67] |
| Hydrothermal | Biomass waste | 14–3.5 | 6 | 205, 260 | Bio imaging | [68] |
| Hydrothermal | *Manilkara zapota fruits* | 5.7, 7.9, 5.2 | 1.9 ± 0.3, 2.9 ± 0.7, 4.5 ± 1.25 | 405, 488, 561 | Bio imaging | [69] |
| Hydrothermal | Broccoli | - | 2–6 | 330–470 | $Ag^+$ sensing | [70] |
| Hydrothermal | Lemon juice | 79 | 4.5 | 540 | Biosensors | [71] |
| Hydrothermal | Cherry tomatoes | 9.7 | 7 | 430 | Biosensors | [72] |
| Microwave-assisted | ND | 26 | ~10 | ND | sensor of $Hg^{2+}$ detection | [73] |
| Microwave-assisted | Cotton linter waste | ND | 10.1 | 420 | Bioimaging | [74] |
| Microwave-assisted | Quince fruit | 8.6 | 4.9 | 450 | Bioimaging | [75] |
| Microwave-assisted | Roasted–Chickpeas | 1.8 | 4.5–10.3 | 435 | Detection of $Fe^{3+}$ | [76] |
| Pyrolysis | Chia seeds | ND | 4 | ND | Sensors | [77] |
| Pyrolysis | Finger millet ragi | ND | 6 | ND | Biosensor | [78] |
| Pyrolysis | Mango | 18.2 | 6 | 525 | Biosensor | [79] |

ND: Not defined.

### 3.1. Hydrothermal/Solvothermal Process

Hydrothermal or solvothermal carbonization strategy is an environmentally friendly, cheap, and nontoxic approach involved in the fabrication of a novel variety of carbon-based substances from different starting materials [65]. Hydrothermal carbonization process has been reported for the synthesis of self-passivated fluorescent CQD in one step consuming several reagents such as acids (ascorbic, citric, and gelatin), animal products (cow milk, bovine serum albumin, and egg albumin), grass, chitosan, food caramels, coffee seeds, orange juice, banana, honey, soy milk, watermelon peels, cellulose, starch, pomelo peel, and paper ash as carbon source. Ideally, a solution of the organic precursor is reacted and sealed in a hydrothermal reactor using a high temperature and the groups attached with CQDs reflect the importance of characteristic fluorescence [80–82]. On the other hand, solvothermal carbonization followed by organic solvent extraction is a well-known technique to fabricate CQDs whereas, carbon-yielding compounds were heated in a high boiling point organic solvents, this is then followed by extraction and concentration procedure [83]. Different natural sources including plants are used in the green synthesis of water-soluble fluorescent carbon dots through hydrothermal/solvothermal treatment in a single step. For example: GQDs (~2.25 to 3.50 nm) have been synthesized using starch by green and one-pot hydrothermal process. The reaction was initiated by hydroxylation then ring-closure condensation and produced hydrophilic GQDs with significant photoluminescence emission and low toxicity. A previous study has been prepared a

high yield of GQDs (44.3%) through the facile hydrothermal method from glucose. The product showed green photoluminescence and excitation-independent photoluminescence emission features [84]. A one-step hydrothermal treatment of natural wastes from *Ananas comosus* and *Citrofortunella microcarpa* was also involved in the green synthesis of CQD [85]. Citrus lemon juice was used as well in the preparation of fluorescent CQDs (~2 to 10 nm) via hydrothermal strategy [86]. It exposed great photoluminescence of 10.2% quantum yield; the photoluminescence intensity was PH-dependent when the maximum intensity appeared at six and effectively applied in the cell. A similar study has been used orange waste peels in the fabrication of amorphous fluorescent CQDs through the hydrothermal carbonization method at mild conditions (180 °C) [87]. It is composed of ZnO and employed in photocatalytic performance using naphthol blue-black azo dye and the superior photocatalytic degradation was distinguished. Other biocompatible and photostable CQDs (~2.0 to 6.0 nm) were produced from by-products in one-pot of hydrothermal action of the biorefinery process (Figure 5) [88]. The significant components of by-products were the degradation products (auto hydrolyzate) of biomass pretreated by autohydrolysis. The quantum yield of blue-green CQDs was about 13% that showed good stability of fluorescence performance, high resistance to photobleaching, and temperature change.

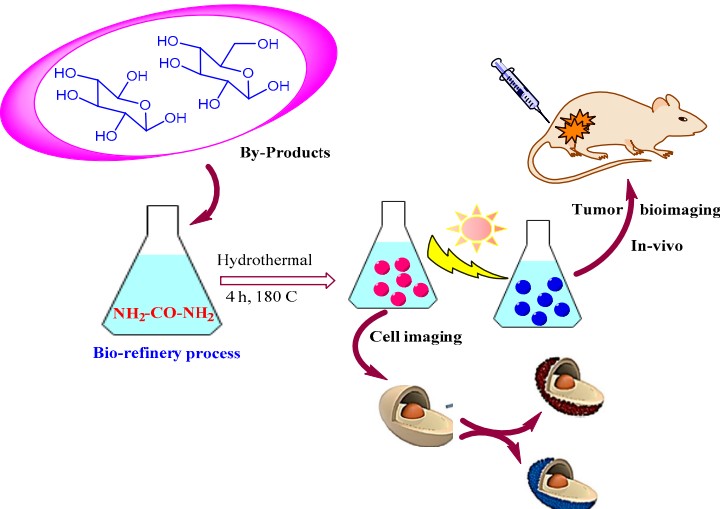

**Figure 5.** Eco-friendly fabrication of CQDs from some of the biorefinery by-products.

On the other hand, cyanobacteria were employed in the fabrication of cost-effective and water-soluble CQDs via a simple hydrothermal approach (Figure 6) [89]. The possible mechanism was suggested in Figure 7. Cyanobacteria are reached by proteins and peptidoglycan and it is reasonable to deduce that the cyanobacteria first suffered from hydrolysis under thermal conditions and produced large amounts of amino acid, N-acetylglucosamine acid, and N-acetylmuramic acid. Polymerization occurred during the synthetic process, and the formed soluble polymers were subjected to carbonization, thus facilitating the formation and growth of carbon cores. Further, since complex compounds took part in the synthetic reaction, the surfaces of the CQDs were more likely to attach multiple functions (Figures 6 and 7) [89]. It was mono-dispersed about 2.48 nm with a quantum yield of 9.24% and characterized by excitation-dependent emission behavior. The cyanobacteria that applied in CQDs did not expose photobleaching under long-time ultraviolet irradiation however displayed great photostability under pH and salinity.

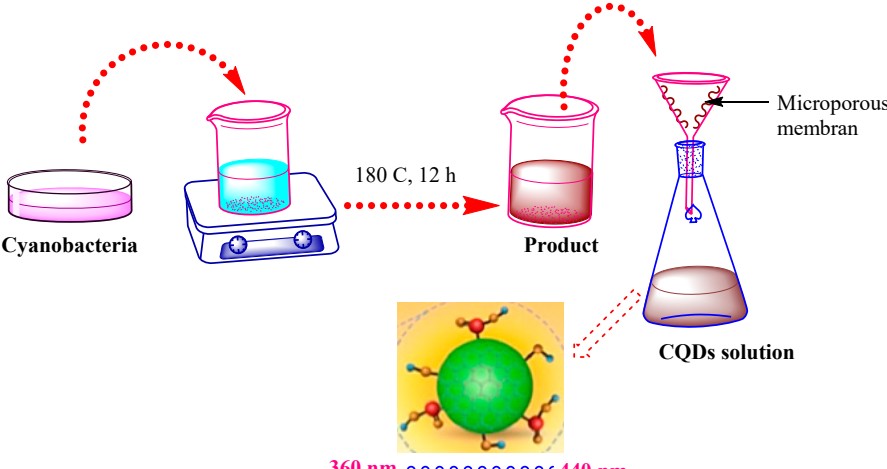

**Figure 6.** Specific synthesis of CQDs from cyanobacteria species.

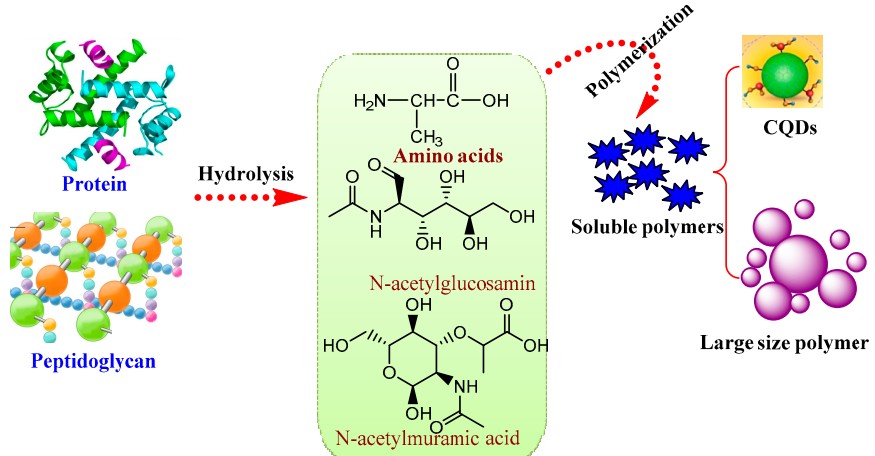

**Figure 7.** Cyanobacteria-derived CQDs possible preparation.

Additionally, coffee grounds were involved in the fabrication of remarkably fluorescent GQDs through an approach that requested hydrazine hydrate-assisted hydrothermal cutting then functionalization by polyethyleneimine. The polyethyleneimine- functionalized GQDs enriched the band-edge photoluminescence with single exponential decay [90]. Further, novel CQDs were recently fabricated via hydrothermal carbonization from renewable chitosan and biocompatible amino acids to produce N-doped chitosan-based CQDs (Figure 8) [91]. The biocompatibility of producing quantum dots was documented, which revealed luminescence in the visible region. It was observed that the quantum yield was influenced by modifications with chemical reagents; glutamic acid and lysine (7.4%, 11.5%), respectively; however, amino acid functionalization did not show a remarkable effect on fluorescence properties. In addition, chitosan was applied in the synthesis of spherical CQDs with two-dimensional structures which demonstrated inhibition corrosion of BIS 2062 carbon steel and also rich in C–O and C = O groups [92].

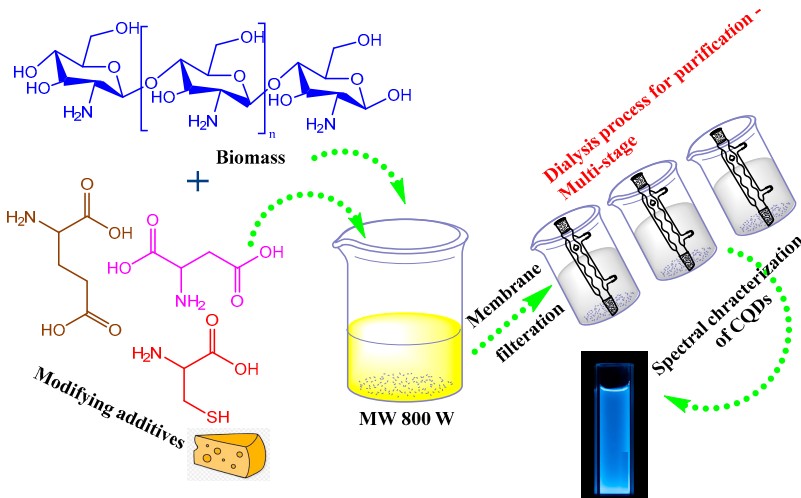

**Figure 8.** N-doped CQDs green synthesis from biopolymeric chitosan.

### 3.2. Microwave-Assisted Heating

Microwave-assisted is an economical and quick procedure that can be used to obtain CQDs, especially when compared to electrochemical and hydrothermal methods. This technology is considered one of the hot areas of research that rapidly developing and utilized microwave rather than direct heating [93]. Carbon-based materials can interact with microwaves significantly. This feature enhanced the technique for producing effectual and localized heating, so the carbonization procedures become favorable and simplify the emergence of distinct morphology of the nanostructures. Several studies have investigated the green synthesis of quantum dots by this technique. For example, an efficient and controlled synthetic approach of carbon dots has been reported using branched polyethyleneimine and citric acid which are employed to change the internal structure [94]. Such a system supports the versatility of carbon dots that could be developed easily by facile fabrication methods between catalytic properties and photoluminescent. More recently, spherical GQDs have been papered through a green approach using cow's milk via a one-pot microwave-assisted heating method. It was observed that the photoluminescence properties were affected by ionic strength and healing time [95]. Therefore, GQDs were biocompatible with the L929 cell line; however, the complex exhibited important cytotoxic activity against several cancer cells. Moreover, a safe synthetic method was investigated for the fabrication of green luminescent graphitic carbon nitride quantum dots coating with sulfur and oxygen and treatment of citric acid and thiourea in the microwave. The luminescence performance was pH-dependent, and the wavelength was excited in the visible region. The result appeared to high quantum yield (31.67%), suitable biocompatibility, and prevent interference in the medium of high ionic strength [96]. Additionally, a recent study investigated the green preparation of CQDs from *Vaccinium Meridionale* Swartz extract through microwave-assisted carbonization. The method was very distinguished in obtaining large amounts of CQDs with concentrations higher than a mass fraction of 80% in only 5 min. The TGA analysis induced that the producing quantum dots (size 30 nm in diameter) showed high thermal resistance even in an atmosphere consisting of air until 300 °C [97].

### 3.3. Pyrolysis

Thermal decomposition has been known as the preferred method for the production of carbon dots which is conducted by pyrolysis or carbonizing the carbon precursors at the increased temperatures [18]. Advantages of the above procedure include simplified operations, solvent-free approaches, wider precursor tolerance, shorter reaction duration, inexpensiveness, and scalable generation. Further, optical features of carbon dots are optimized through alterations in main factors like the reaction temperatures, duration,

and reaction mix pH [98]. A high fluoresce CQDs (~6 nm) were green synthesized via the pyrolysis of *Eleusine coracana* where $Cu^{2+}$ strongly quenched the fluorescent capacity of CQDs comparing with other metal ions. The metal ions of $Cu^{2+}$ preferred to adsorb on the surface of CQDs through π-bond of aromatic CC, while other divalent metals desired σ-bond with CQDs [78]. In addition, mono-dispersed CQDs were prepared by a one-step thermal decomposition strategy from fennel seeds of *Foeniculum vulgare* as displayed in Figure 9 [99]. The product displayed significant photostability, colloidal, stability against pH changes, and no need for additional surface passivation step to develop fluorescence. These particles revealed significant excitation- independent emission and photoluminescence activity as well. Some factors are affecting the pyrolysis process such as time of reaction and temperature that improve our vision about the formation of CQDs.

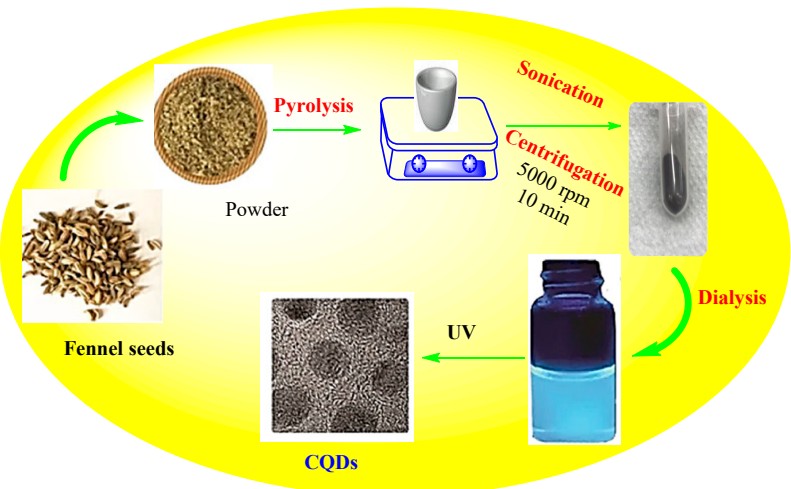

**Figure 9.** Fennel seeds CQDs by pyrolysis technique.

On the other hand, silver nanoparticles supported by GQDs and silica are engineered and formulated via a greener photochemical strategy and electrostatic deposition technology affording highly active surface-enhanced for the Raman scattering layer. The prepared aqueous solution of GQDs that was prepared during the reaction was employed as a solvent and a reducing agent to cause in-situ silver-GQDs composite under UV irradiation conditions. Silicon dioxide was used to collect the composite by electrophoresis deposition system (Figure 10) [100]. GQDs could use as important sites to illuminate the signals of Raman scattering, due to the suitable size (~1 to 4 nm) and the good distribution between gaps and Ag nanoparticles. Because of the improving adsorption of Rhodamine 6G particles via π-π stacking, magnified specific surface area through $SiO_2$ pattern and electrostatic interactions from GQDs, the as-prepared substance revealed significant surface-enhanced Raman scattering signal with significant reproducibility, the detection limit of Rhodamine 6G was increased up tar pitch, has a special structure containing an aromatic nucleus with numerous side chains that bond on this graphene-like nucleus, that similar to the structure of GQDs.

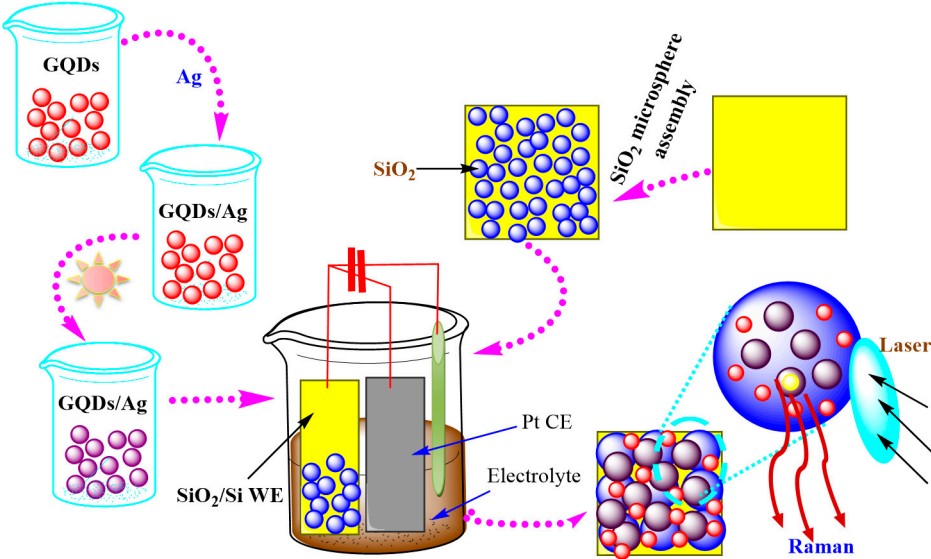

**Figure 10.** Formation of Ag-GQDs/SiO$_2$/Si substrate.

## 4. Biomedical and Biotechnological Applications

Since the last decades, carbon dots are the new fellow of the carbon family with size less than 10 nm that gained much interest of researchers due to their exclusive properties, such as facile and inexpensive synthetic ways, low toxicity, availability of surface modification, outstanding photoluminescence, and excellent water solubility [24]. Due to these unique features, carbon dots have been widely applied in several types of scientific fields. Phosphorescence and fluorescence are the most imperative phenomena found in carbon dots which enhance their uses for in vivo and in vitro biosensing and bioimaging. It could also be used in drug delivery, photocatalytic reactions, photodynamic, and photothermal therapies. However, some problems are related to employing carbon dots with metal ions that are usually toxic and environmentally hazardous whereas traditional carbon dots are non-toxic and much safer, affording their good biological and environmental compatibility [101,102]. Sensor and bioimaging applications of CQDs produced by green chemistry methods are classified and analyzed according to the synthesis methods of CQDs and summarized in Table 1.

### 4.1. Cancer Therapy and Drug Delivery

CQDs have a strong ability to act as drug and gene carriers due to their biocompatibility, photoluminescence, and non-toxicity, while the very small size and large surface area permit fast cellular uptake with little effect on the activity of the drug [103,104]. For instance, the antitumor drug (doxorubicin) was successfully loaded on the surface of the composite (arginine-glycine-aspartic acid-GQDs) which was applied in drug delivery and targeted imaging. The significant fluorescent-GQDs able to analyze the cellular uptake at a definite time then the doxorubicin drug was released. The drug release was found to be influenced by pH and the interaction between doxorubicin and GQDs through hydrogen bonds. Doxorubicin conjugated with GQDs displayed potent cytotoxic activity against U251 glioma cells comparing to free doxorubicin. By investigation of the cellular uptake, it was found that some of GQDs and the tumor drug doxorubicin were penetrated until reaching the cell nucleus after incubation (~16 h). This behavior increase capability of cytotoxicity of doxorubicin [105]. An excellent smart stimuli-response drug delivery system has been studied that consist of CQDs coated with alginate beads and garlic crude [106]. By comparing the presence of garlic extract loaded on the surface of alginate beads in coated carbon dots and uncoated alginate beads was found the amount 60% higher. The system was found to be pH-dependent controlled drug release and caused therapeutic effectiveness, stimulatingly, based on the amount of pathogen that existed in the target. Additionally, folic

acid attached with GQDs was also examined that used in doxorubicin loading, detection of real-time of cellular uptake consequently drug release. This precisely fabricated nanostructured was immediately assimilated by HeLa cells through receptor-mediated endocytosis, while release and accumulation of doxorubicin continued. The in vitro studies confirmed the effective and significant cytotoxicity of the synthesized nano assembly targeted HeLa cells, but the effect decreased for the non-target cells. Liu et al. [107], also described the vector of insoluble aromatic drug SN38 via a PEGylated nanographene oxide that having a size between 5 and 50 nm. The delivered cancer-killing drug was found to be 1000-fold more potent than the approved drug by the FDA used in the treatment of colon cancer. Furthermore, CQDs coupled with Au nanoparticles for an assembly, then conjugated with PEI–pDNA for delivering DNA to cells. Fluorescence of CQDs could be quenched by Au nanoparticles; thus, pDNA release could be probed by the recovery of the fluorescence signals. The experimental results showed that the assembly entered into the cells with the CQDs located in the cell cytoplasm and the pDNA released entered the cell nuclei, achieving critical transfection efficiency (Figure 11) [108].

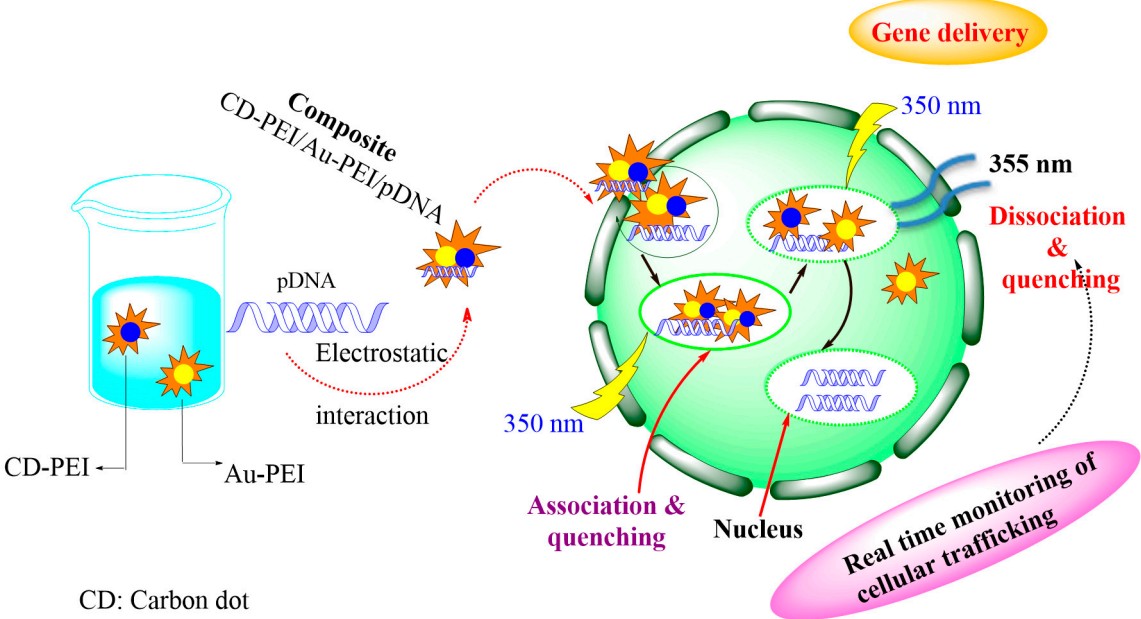

**Figure 11.** Schematic chart of gene delivery of CQDs.

The pancreatic cancers (MiaPaCa-2 cells) were also investigated by applied GQDs with biodegradable charged polyester vectors [109]. The producing substance was utilized as nanocarriers for loading of doxorubicin and quite interfering ribonucleic acids. It also exhibits remarkable physiological stability, excellent K-ras down-regulation effect, and operative bioactivity inhibitions. Furthermore, the cells could be destructed using laser light that generates heating for the nano complexes; consequently, the photothermal caused cell death. The laser caused the firing of payloads from the formed nanostructured composites, and this active firing function greatly enhanced their anticancer activity. From the relevant studies, these composites could be used as remarkable tools as a drug carrier for in vivo examinations. In addition, gene delivery has been proved significantly with positively charged carbon dots that may be attached with plasmid DNA and therapeutic plasmid was professionally transferred into the cells and low toxicity [110]. It was found that polyethyleneimine and hyaluronate–functionalized CQDs were internalized readily into the cytoplasm of cancer via hyaluronate-receptor-mediated endocytosis. They displayed outstanding gene condensation compatibility through electrostatic attraction and protective capability by avoiding nuclease degradation [22].

### 4.2. Imaging and Bioimaging

Live cell bioimaging is becoming an increasingly popular tool for elucidation of biological mechanisms and is instrumental in unraveling the dynamics and functions of many cellular processes. Bioimaging is a method for imaging and direct visualization of biological processes in real-time which is often used to gain information on the 3D structure of the observed specimen from the outside, i.e., without physical interference [111]. The optical features of quantum dots with carbon basis are key factors of the practical application of these compounds because they release heavy fluorescence and have lower cytotoxicity and high biocompatibility that could be used in bioimaging [112]. CQDs and GQDs are widely used in bioimaging applications that are involved in cell imaging as displayed in Table 2.

**Table 2.** Recent bioimaging application of CQDs and GQDs.

| Cell Line | Imaging Position | Quantum Dots Conc. | Color | Ref. |
|---|---|---|---|---|
| L929 fibroblasts | Membrane, cytoplasm | 0.30 mg/mL | ND | [68] |
| HeLa | Membrane, cytoplasm | 10 μg/mL | Blue, green, yellow | [69] |
| Nematodes | ND | 100 μg/mL | Blue, green, red | [65] |
| H2452, HUVEC | ND | 50 μL/mL, 100 μL/mL | Blue | [74] |
| HT-129 | ND | ND | Blue | [75] |
| HEK-293 cells | Cell membrane | 40 μg/mL | Multi-color | [113] |
| A549 MCF-7 | Cell cytoplasm | 25 μg/mL | Yellow | [114] |
| HeLa | Cell nucleus | 0.01 mg/mL | Green | [115] |
| MCF-7 | Cell membrane, cytoplasm, nucleus | 100 μg/mL | Green | [116] |
| MDA-MB231 | Cell | 0.1 mg/mL | Red | [117] |
| MC3T3 | Cell cytoplasm | 2.5 mg/mL | Bright green or blue | [118] |

Therefore, it has become suitable for clinical and biological imaging and related diagnostic and therapeutic areas, such as phototherapies and diagnostic cancer imaging. For example, 200 μL of CQDs conjugated with wheat straw (0.2 μg mL$^{-1}$) was injected through the tail vein of the mouse and the optical imaging was investigated [88]. The utilization of CQDs as fluorescent labels in imaging different cells has been investigated [119]. CQDs are successfully applied in the field of bioimaging due to their remarkable properties including, low toxicity, eco-friendly and fewer side effects, have strong ability to soluble in water, and visible-to-near infrared (NIR) emission properties [120]. Different cell lines have been imaged by CQDs such as Ehrlich ascites carcinoma cells, HepG2 cells, Escherichia **coli** *(E. coli)*, HeLa cells, human lung cancer (A549), and NIH-3T3 fibroblast cells [120–124]. Nescafe instant coffee was involved in yielding CQDs (quantum yield ~5.5%) with a size of 4.4 nm [125]. Coffee-derived CQDs have been used to image carcinoma cell lines and small guppy fish without functionalization. Human breast cancer MCF-7 can be detected and imaging through carbon dots passivated with PPEI-EI for two-photon luminescence microscopy [80]. Bright photoluminescence was induced in the cytoplasm and cell membrane at 37 °C after 2 h incubation whereas the cellular uptake of carbon dots was temperature-dependent with no internationalization perceived at 4 °C. Two photons as turn-on fluorescent probes were applied in bioimaging for the $H_2S$ in tissues and cells [126]. It is worthy to mention that CQDs were modified with Cu(II) complex that quenched quantum dots fluorescence and then, competitive copper sulfide formation caused fluorescence dequenching. A previous study prepared a large quantum yield of bright green-GQDs (11.4%) that are highly soluble in $H_2O$ and several organic solvents without any change and excellent photoluminescence [127]. Furthermore, yellow-green-photoluminescent GQDs (almost 10 nm) were fabricated by strong oxidation of graphite; the formed substance

displayed low toxicity, excellent photostability, and good solubility. The fluorescence quantum yields were estimated by 7% and have been applied in cell bioimaging [128]. The bioimaging applications still faced some problems that required further investigations. Several reports have previously investigated the fabrication of GQDs with various emission wavelengths between ultraviolet to near-infrared, but the producing quantum yield was found to be lower than conventional semiconductor quantum dots. Hence the development of quantum yield became an urgent request. GQDs with strong red-near infrared emission could be used as appropriate nano-probes in bioimaging [83,129]. Nano-probes with several functions could be an answer for industrial and environmental challenges in the field of imaging and bioimaging. GQDs can be applied in the field of therapy and imaging for the same purposes due to their optical and radioactive properties. Different challenges must be considered in the fabrication of GQDs-based nano-probes which are used in optical imaging concurrently with magnetic resonance imaging, and evaluation of computed tomography. Very few reports have rarely investigated the imaging of in-vivo targeted tumors through GQDs. Cancer diagnosis in young animals involves great accumulations in the tumor tissues. The role of antibodies/or peptides-GQDs in imaging of cancer target needs further investigation. Several issues related to GQDs are still faced drawbacks and problems that need to be solved and put under deep studies for example; excitation of multi-photons, therapy of brain gene, applications and innovation in neurobehavior, and penetration of brain barrier in blood vessels [130–132]. Additional studies are still requested to evaluate the cytotoxic effect of GQDs with some factors including different morphologies, sizes, and surface coating [116]. Cost-effective and eco-friendly methods have been used to prepare GQDs improved with polyethyleneimine or (3-carboxyl) phenyl bromide phosphine that produced on a large scale, whereas GQDs-polyethyleneimine was synthesized by a simple hydrothermal process. In addition, GQDs-polyethyleneimine was conjugated with (3-carboxyl) phenyl bromide phosphine that attached by an amide linkage. The average sizes of the formed two composites were estimated by 3.75 and 3.25 nm, respectively. Both fabricated substances displayed low cytotoxicity, important optical feature, and showed selectivity to image mitochondria or cell nucleus. This is reflecting the vital role of GQDs in bioimaging particularly cell nucleus and mitochondria imaging in vitro and in vivo for diagnosis and therapy [133].

### 4.3. Anti-Microbial Activity

CQDs can interact successfully with different viruses and retard infection [134]. For example, CQDs attached with amino groups or boronic acid could affect the entry of the herpes simplex virus type 1. Hydrothermal carbonization strategy was employed in the fabrication of CQDs by using 4-aminophenyl boronic acid hydrochloride; the product was found to be effective against herpes simplex virus type 1. Other material was fabricated from phenylboronic acid but did not show any activity at the applied concentration. These CQDs can be used against one of the most relevant pathogenic human infections today (coronavirus). Mechanistically, it may be due to the human coronavirus-229E entrance inhibition, caused by the interaction of the boronic acid functions of CQDs with the HCoV-229E S protein through pseudo-lectin-based interactions. The results support scientists to replace the current applied antiviral substances (e.g., interferons and ribavirin). These agents displayed many side effects, such as losing memory in the short-term, inhibition in the function of decision-making, confusion, and extrapyramidal effects. More effort is highly recommended to examine deeply the clinical trials for such materials as suggested candidate for therapy and considered one way to challenge the difficult and life-threatening diseases [135]. CQDs have been used as an antimicrobial against different types of bacteria including, *Pseudomonas aeruginosa, E. coli,* and *Staphylococcus aureus*, also used in the imaging of these microbes. CQDs work to identify the gram type of bacteria, evaluation of microbial viability, and image biofilm [136,137]. It worth mentioning that the layers of CQDs play a significant role in killing bacteria because it is used basically as a carrier for conventional disinfection agents, while the label of fluorescence is involved for analyzing

the dead bacterial cells. The quantum dots choose to interact with Gram-negative then adsorbed on the surface after then the fluorescence emission increased clearly. The mechanism of interaction occurs by verifying the balance of surface charge and CQDs inserted to the surface from long alkyl chains, hence this action caused destructive cell wall and bacterial inactivation. This differentiation strategy is distinguished by simplicity, fast, and occurs easily [131]. The preparation process of carbon dots and their application for selectively imaging and killing Gram-positive bacteria is represented in Figure 12. Glycerol was used in fabrication process as a cheap source of carbon source and can afford hydroxyl groups for the final product of carbon dots to enhance its good water dispersibility. The organosilane molecule Si-QAC, containing a quaternary ammonium group and a long hydrocarbon chain, was used as the carbon source and surface passivation agent. The quaternary ammonium group and the long hydrocarbon chain of Si-QAC are crucial for endowing the final carbon dots product with excellent antibacterial activity since these two moieties can interact with the bacterial surface with electrostatic and hydrophobic interactions.

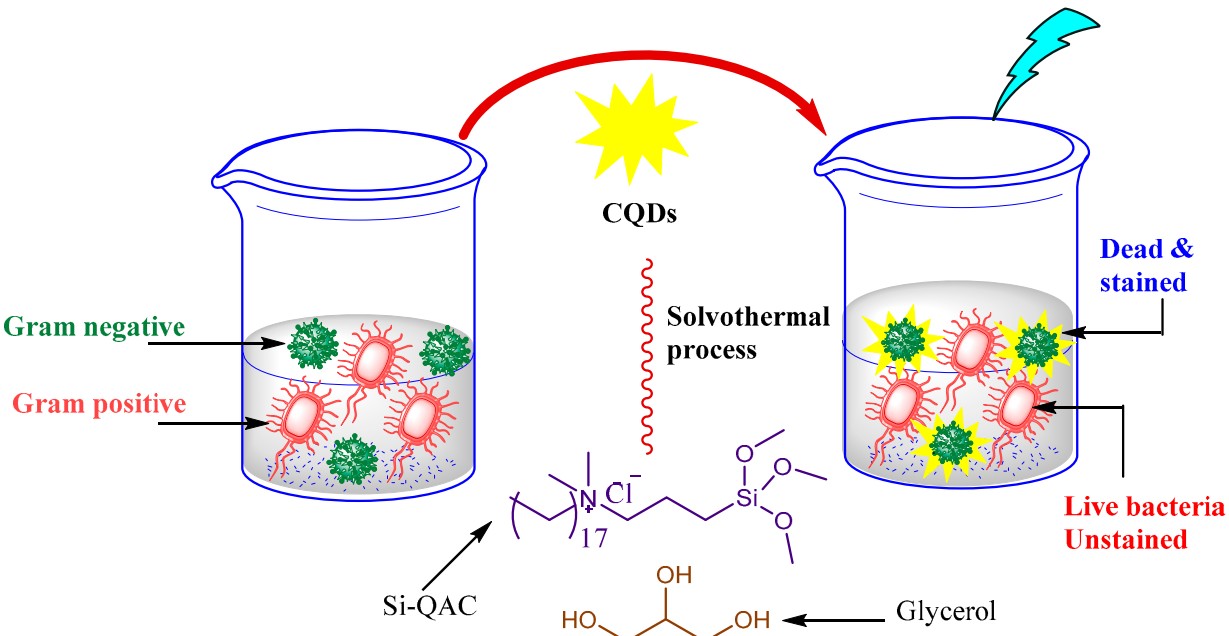

**Figure 12.** Quaternization of carbon dots formation and selective imaging potentiality against Gram-positive bacteria.

### 4.4. Sensors and Biosensors

In literature, a lot of papers are designed for water treatment [138–140], and there are several types of sensors fabricated from CQDs which have been involved in the identification of specific targets such as glucose [141], DNA [142], heavy metals [143], phosphate [144], proteins [145], $H_2O_2$ [146], and nitrite [147]. For instance, an electrochemical carbonization strategy of urea and sodium citrate was used in the preparation of CQDs that revealed high selectivity and sensitivity towards $Hg^{2+}$ and photoluminescence emission whereas the detection limits were 3.3 nM and 0.5 nM, respectively [148,149]. Several other heavy metals were successfully detected as well including; $Sn^{2+}$, $Cr^{6+}$, $Fe^{3+}$, $Mn^{2+}$, $Pb^{2+}$, and $Cu^{2+}$ as tabulated in Table 3 [150–152].

**Table 3.** Different examples of biosensor application of CQDs and GQDs.

| Analyte | LOD | Range | $\lambda_{em}$ Max | Quantum Yield (%) | Size Range (nm) | Ref. |
|---|---|---|---|---|---|---|
| $Zn^{2+}$ | 5.4 μM | 0–125 μM, 125–200 μM | 270, 283 | 21.3 | 3.7 | [66] |
| $Fe^{3+}$ | 0.073 μM | 0.1–0.9 μM | 445, 435, 43, 435 | 4.3–8.2 | 1.3 and 4.9 | [67] |
| $Ag^+$ | 0.5μM | 0–600 μM | 330–470 | - | 2–6 | [70] |
| $V^{5+}$ | 3.2 ppm | 0–100 ppm | 540 | 79 | 4.5 | [71] |
| Trifluralin | 0.5 nM | 0.050–200 μM | 430 | 9.7 | 7 | [72] |
| $Hg^{2+}$ | 1.78 μM | 5–50 μM | ND | 26 | ~10 | [73] |
| $As^{3+}$ | 0.02 μg/mL | 0.1–2 μg/mL | 450 | 8.6 | 4.9 | [75] |
| $Fe^{3+}$ | 2.8, 8.2 μM | 11.3, 37.5 μM | 435 | 1.8 | 4.5–10.3 | [76] |
| Hydrazine | 39.7 μM | 125–1125 μM | ND | ND | 4 | [77] |
| $Cu^{2+}$ | 10 nM | 0–100 μM | ND | ND | 6 | [78] |
| $Fe^{2+}$ | 0.62 ppm | ND | 525 | 18.2 | 6 | [79] |
| $Fe^{3+}$ | 0.21 μM | 0–300 μM | 450 | 22 | 6 | [153] |
| $Hg^{2+}$ | 2.3 nM | 5–100 μM | 428 | 6.4 | 2.8 | [154] |
| $Hg^{2+}$ | 2.6 μM | 10–100 μM | 420 | 9.6 | 8 | [155] |
| $Fe^{3+}$ | 0.56 μM | 50–350 μM | 493 | 11.2 | 3 | [156] |
| $Fe^{3+}$ | 0.5 μM | 0–1.7 mM | 450 | 8 | 2.8 | [157] |

It was found that the produced CQDs were influenced by excitation behavior and the quantum yield was significant (~46.6%). The excellent sensitivity supports their utilization in turn-off $Hg^{2+}$ detection with a minimum limit of detection as low as 6 nM in the dynamic range from 0 to 0.1 μM. It is also used as a turn-on sensor to detect glutathione with high selectivity [158]. Carbon dots modified by boronic acid have been investigated to determine non-enzymatic blood glucose. The glucose level has appeared in the range between 9 and 900 μM and the detection limit was 1.5 μM. The results of this method were consistent with the values determined by a commercial blood glucose monitor [159]. CQDs based fluorescence turn-on sensors were instructed to monitor $H_2O_2$ in an aqueous medium. This is occurred by the mechanism of photo-induced electron transfer whereas the sensor displayed proper sensitivity and selectivity with a detection limit of 84 nM [160]. In addition, CQDs doped with N and S have been prepared by thermal reaction between ethylenediamine, ammonium persulfate, and glucose. The producing material revealed bright blue emission with a strong fluorescent quantum yield of 21.6%. The product showed significant characteristics comparing with carbon dots only such as high stability, easily soluble in water, and uniform morphology. The fluorescence of the produced carbon nano-dots could be remarkably quenched by methotrexate. This may be occurred by fluorescence resonance energy transfer between methotrexate and carbon nano-dots. The significant selectivity supports the effective recognition of methotrexate (more than 50.0 μM) with a low detection limit of 0.33 nM. It also could be used for hands-on identification of methotrexate in human serum [161].

On the other hand, it is worth mentioning that A few numbers of research spot the light on the utilization of CQDs and GQDs in biosensing so further and deep investigations are highly appreciated. Shi et al., [162] have been applied GQDs with Au nanoparticles to enhance the fluorescence resonance energy transfer biosensors and utilized in specific detection of a particular gene sequence in *Staphylococcus aureus*. Whereas Zhang et al. [163], modified GQDs and employed them in biosensors fabrications that aimed at the detection of microRNA. The study reported that changes in fluorescent intensity caused a signal detecting microRNA with appropriate discrimination capabilities in the range from 0.1

to 200 nM. CQDs functionalized with amino groups were also applied in biosensors and for selective identification of hyaluronidase [164]. In addition, CQDs and GQDs could be used in the manufacture of immuno-sensor. These types of sensors can be fabricated from 8-hydroxy-22′-deoxyguanosine while CQDs doped with $Au/SiO_2$coreshell nanoparticles were immobilized on the surface of the platinum electrode [19].

## 5. Limitations and Future Prospective

In recent decades, carbon dots have gone through a great revolution and have become one of the key areas for humans. However, certain gaps and points remain secret. The origin of fluorescence emission is widely discussed and further research is needed [165]. The scientists have not defined yet the role and effect of carbon dots structures on their properties; but this limitation did not prevent researchers from deep investigation for synthesis and applications particularly in the biological sciences [166,167]. Despite the several applications of CQDs in the field of biomedical, their effect on the bloodstream is still unclear and further investigations are highly recommended. Although quantum dots have been investigated in bioimaging, serious health problems and environmental concerns limit their bio-applications in this area owing to the presence of heavy metals [98]. Further, the area of biosensing applications still an enigma as few numbers of studies spot the light on the utilization of CQDs and GQDs in this field so, more examinations may reply to many questions and revealed new applications [168]. CQDs can be easily synthesized from several natural carbon sources but these substances lack the homogeneity and the actual purity for producing homogenous CQDs [169,170]. The carbonaceous aggregation considered another problem arising from the carbonization of CQDs during different synthetic methods, e.g., pyrolysis, and electrochemical. As well, surface features that are critical for solubility and specific applications, may be altered during synthetic methods or post-treatment [171]. Moreover, the traditional green methods employed in the fabrication of GQDs still faced many problems. It usually required strong acids or other organic solvents and may be complicated post-processes [172]. So, there is an urgent demand to develop eco-friendly techniques that depend on natural renewable sources and easy separation.

## 6. Conclusions

In the last years, carbon dots received tremendous attention and interest as an excellent candidate comparing with the common semiconductor quantum dots because of their incomparable and distinctive characteristics. The current review covers information about the current synthetic strategies of carbon dots with emphasis on the green synthetic approach which becomes more attractive and matching with the rule of eco-friendly chemistry. Despite several advantages and the unique properties of quantum dots, some drawbacks and limitations remain. This review focuses on the advantages of carbon dots in several applications, particularly in biomedical aspects.

**Author Contributions:** The idea and contents, M.M.T., R.M.E.-S., and M.F.E.; Investigation and literature survey, M.F.E.; Software, R.M.E.-S.; Validation, B.M.A., Writing—original draft, R.M.E.-S.; Writing—review and editing, M.F.F., Revision, B.M.A. and K.N.M. All authors have read and agreed to the published version of the manuscript.

**Funding:** This research was funded by Deputyship for Research and Innovation, "Ministry of Education" in Saudi Arabia through Project no. (IFKSURP-21).

**Institutional Review Board Statement:** Not applicable.

**Informed Consent Statement:** Not applicable.

**Data Availability Statement:** Not applicable.

**Acknowledgments:** The authors extend their appreciation to the Deputyship for Research & Innovation, "Ministry of Education" in Saudi Arabia for funding this research work through the project number IFKSURP-21.

**Conflicts of Interest:** On behalf of all authors, the corresponding author states that there is no conflict of interest.

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
