# Peer review of "Recent Developments in Carbon Quantum Dots: Properties, Fabrication Techniques, and Bio-Applications"

_processes, doi:10.3390/pr9020388_

Round 1
Reviewer 1 Report
Submitted manuscript entitled „ New trends carbon quantum dot: ecofriendly process, properties, and biomedical applications” attempted to report recent developments in carbon quantum dots (CQDs) and graphene quantum dots (GQDs). The subject of the review is interesting but the text is difficult to the readers because of poor presentation and organization of the content. The manuscript has been badly prepared and contains numerous errors, unclear statements and mistakes. English is unacceptable and even the title needs revision. Although the manuscript was divided into several sections including syntheses, characterization, and applications, all informations were not properly arranged. Section related to syntheses refers also applications, sections devoted to applications contain also data on QDs preparation. Concluding, the review is superficial and one can suspect that Authors lack expertise in the reviewed field. Below are general drawbacks of submitted paper.
- - CQDs and GQDs should be clearly distinguished and reviewed separately
- - Table reporting new synthesis protocols of quantum dots should be presented and critically discussed
- - Table showing selected applications of (C,G)QDs in sensing and biosensing should be presented
- - Table reporting selected applications in bioimaging and drug delivery should be presented
- - Several important review papers were not cited (e.g., J Mater Sci,51, 4728–4738(2016); Microchim Acta 183, 519–542(2016); 184, 1899–1914(2017); Polymers, 9(2), 67 (2017); RSC Adv., 10, 15406–15429 (2020)
Other issues
- - some references are missing or cited in different way. For example, the fragment on page 9 (line 265-279) and Fig. 9 lacks reference citation. Generally references are cited as [XX] but on page 11 one can find: (Liu et al., 2008).
- - There are two Figs 8
- - 10 is cited incorrectly (“The delivered cancer-killing drug was found to be 1000 fold more potent than the approved drug by FDA used in the treatment of colon cancer (Figure 10).”) . The gene delivery system shown in Fig.10 has not been explained in the text.
- - On page 14 Authors wrote: “Methotrexate could be quenched through carbon nano-dots fluorescence. This may be occurred by flu-439 orescence resonance energy transfer between methotrexate and carbon nano-dots.” This is false statement. Correctly, methotrexate quenched fluorescence of CQDs!
- - Other example of a brief and incomplete description of reported reference one can find also on page 14 line 465: “Two photons as turn-on fluorescent probe were applied in bioimaging for the H2S in tissues and cells [115]”. This sentence is true but important information is missing (QDs was modified with Cu(II) complex that quenched QDs fluorescence and next, competitive copper sulphide formation caused fluorescence dequenching).
- - Page 4 line 100: “The color of the fabricated carbon dots was changed from ultraviolet to red,…”. Ultraviolet is not a color.
I do not recommend this manuscript for publication in Processes.
Author Response
Reviewer#1
Submitted manuscript entitled „ New trends carbon quantum dot: ecofriendly process, properties, and biomedical applications” attempted to report recent developments in carbon quantum dots (CQDs) and graphene quantum dots (GQDs). The subject of the review is interesting but the text is difficult to the readers because of poor presentation and organization of the content. The manuscript has been badly prepared and contains numerous errors, unclear statements and mistakes. English is unacceptable and even the title needs revision. Although the manuscript was divided into several sections including syntheses, characterization, and applications, all information’s were not properly arranged.
Section related to syntheses also refers applications, sections devoted to applications also contain data on QDs preparation.
Author’s reply: Thank you very much for your valuable comments and we accordingly changed the text and removed the unrelated parts. Otherwise, unclear statements and mistakes were manipulated. The grammar checker was assessed in the revised version.
Concluding, the review is superficial, and one can suspect that Authors lack expertise in the reviewed field. Below are general drawbacks of submitted paper.
- CQDs and GQDs should be clearly distinguished and reviewed separately
Author’s reply: Thank you for your advice. In the present review, the main goal is the green synthesis for CQDs and GQDs which based on carbon source, so we have to correlate between CQDs and GQDs, and in my opinion, it is hard to separate between them and some parts in the review we separate between them in their proper positions.
- Table reporting new synthesis protocols of quantum dots should be presented and critically discussed
Author’s reply: Thank you for your suggestions. As per your request new table was added to the revised manuscript explained different synthetic approaches, main sources, quantum yield efficiency, size range, etc.
- Table showing selected applications of (C,G)QDs in sensing and biosensing should be presented
Author’s reply: Thank you so much for your suggestions. The table is inserted in the text with mentioned applications.
- Table reporting selected applications in bioimaging and drug delivery should be presented
Author’s reply: Thank you so much for your suggestions. In the same table, the bioimaging applications were added.
- Several important review papers were not cited (e.g., J Mater Sci,51, 4728–4738(2016); Microchim Acta 183, 519–542(2016); 184, 1899–1914(2017); Polymers, 9(2), 67 (2017); RSC Adv., 10, 15406–15429 (2020)
Author’s reply: Thank you for your kind advice. The mentioned references were added to the revised manuscript in their proper positions.
Other issues
- some references are missing or cited in different way. For example, the fragment on page 9 (line 265-279) and Fig. 9 lacks reference citation. Generally references are cited as [XX] but on page 11 one can find: (Liu et al., 2008).
Author’s reply: Thank you for your careful view. The missing reference is cited and formatted according to the journal style.
- There are two Figs 8
Author’s reply: Thank you for your notice, figures are re-numbered correctly.
- 10 is cited incorrectly (“The delivered cancer-killing drug was found to be 1000 fold more potent than the approved drug by FDA used in the treatment of colon cancer (Figure 10).”) . The gene delivery system shown in Fig.10 has not been explained in the text.
Author’s reply: Thanks once more for your kind concern. The figure is explained in the text and cited in the correct position.
- On page 14 Authors wrote: “Methotrexate could be quenched through carbon nano-dots fluorescence. This may be occurred by flu-439 fluorescence resonance energy transfer between methotrexate and carbon nano-dots.” This is false statement. Correctly, methotrexate quenched fluorescence of CQDs!
Author’s reply: Many thanks for your effort. The statement was checked and rewritten.
- Other example of a brief and incomplete description of reported reference one can find also on page 14 line 465: “Two photons as turn-on fluorescent probe were applied in bioimaging for the H2S in tissues and cells [115]”. This sentence is true but important information is missing (QDs was modified with Cu(II) complex that quenched QDs fluorescence and next, competitive copper sulphide formation caused fluorescence dequenching).
Author’s reply: Thank you for your concern. The text is completed and highlighted with red color in the revised version.
- Page 4 line 100: “The color of the fabricated carbon dots was changed from ultraviolet to red,…”. Ultraviolet is not a color.
Author’s reply: Thank you for your observation. The sentence is modified in the final version.

Reviewer 2 Report
This review article discusses the ecofriendly process, properties, and biomedical applications of carbon dots. I found several drawbacks of this article and the authors should consider some revisions for improvement of the article before it can be further proceeded.
- The title should be corrected, there is a grammar mistake. Carbon dots should be written as carbon dots.
- The authors did not mention previous review papers about carbon dots: synthesis, properties, and biomedical applications. What are the new trends in this paper? It is better to make specific years of eco-friendly process and biomedical applications such as within 5 years or 10 years.
- The authors did not mention the gap of this review paper compared to the currently published review paper due to many similar review papers discussed the same topic.
- Authors should focus also on the writing issue including the numbers and chemical formula such as H2O, sp2, NH2, etc. (should be H2O, sp2, NH2). The authors should carefully check one by one.
- The authors mentioned in section 3.1. hydrothermal/solvothermal process, but the authors did not explicitly explain the difference between these two methods. In fact, these two methods are a little bit different. Authors should explore more. Is the solvothermal an eco-friendly method?
- The authors mentioned that only three methods as eco-friendly processes and these methods are bottom-up methods. Authors should discuss how can these methods are eco-friendly compared to the top-down methods such as laser ablation, electrochemical reduction, etc.
- Authors are suggested to make a table for green starting materials used in the synthetic process. There are many new papers that have been published in 2019-2020 where carbon dots have been prepared using new natural and green starting materials by hydrothermal, pyrolysis, and microwave methods and add their references in the table.
- I found a word mistake inside Figure 3, correct it if there is a mistake. Please check Figure 8, I cannot see the full figure in the pdf file.
- In Figure 9, the authors discussed Ag-CQDs/SiO2/Si substrate. What is the purpose to discuss this material? It is better if the authors focus on carbon dots only not on their composites.
- The authors added several biomedical and biotechnological applications including cancer therapy and drug delivery, imaging and bioimaging, anti-microbial activity, and sensors. What is different between imaging and bioimaging. I cannot find it clearly in the text. Here, the addition of “sensors” part is not correlated to the biomedical applications. Authors should focus only on biomedical applications. It is recommended to remove “sensors” part. If the authors want to add “biosensors” part, then the title should change from biomedical applications to bio-applications.
- Please make sure that the carbon dots applied in biomedical applications were synthesized using eco-friendly process. Authors should not randomly collect the references without the relation with eco-friendly synthetic process of carbon dots.
- Overall, the authors are recommended to comprehensively check the grammar mistakes, words, and chemical formula. Besides, the synthesis process of carbon dots should be discussed better and more compared to other parts, due to the manuscript submitted to the “processes” journal. If the authors focus on the writing of “process” part then we can understand the benefit of this article compared to various similar review articles.
Author Response
Reviewer#2
Comments and Suggestions for Authors
This review article discusses the ecofriendly process, properties, and biomedical applications of carbon dots. I found several drawbacks of this article and the authors should consider some revisions for improvement of the article before it can be further proceeded.
The title should be corrected, there is a grammar mistake. Carbon dots should be written as carbon dots.
Author’s reply: Thank you for your careful view. Accordingly, we carefully revised the manuscript and the title was changed to be “Recent developments in carbon quantum dots; properties, fabrication approaches, and biotechnological applications”
The authors did not mention previous review papers about carbon dots: synthesis, properties, and biomedical applications. What are the new trends in this paper? It is better to make specific years of eco-friendly process and biomedical applications such as within 5 years or 10 years.
Author’s reply: Thank you for your notice. The main goal of the present review is to mention the green synthetic procedures of carbon quantum dots, properties, and applications. In the revised version we changed a lot of concepts and different references were added to the revised manuscript. Also, the biomedical applications were separately added in Table 1, with applications in the area of biomedical applications as your request. All changes were marked with red color in the revised manuscript.
The authors did not mention the gap of this review paper compared to the currently published review paper due to many similar review papers discussed the same topic.
Author’s reply: Thank you for your question. Some certain gaps and points remain unclear in carbon quantum dots. The origin of fluorescence emission is widely discussed, and further research is needed. The role of carbon dots structures and how they affect their properties not yet determined, but this limitation does not prevent researchers from conducting extensive research on the synthesis and some applications in the biological sciences, etc.…Limitations section was rewritten to point out the main gaps of general carbon dots and this part was colored with red.
Authors should focus also on the writing issue including the numbers and chemical formula such as H2O, sp2, NH2, etc. (should be H2O, sp2, NH2). The authors should carefully check one by one.
Author’s reply: Thanks again, all the text has revised carefully, and all chemical formula was rewritten by a true manner.
The authors mentioned in section 3.1. hydrothermal/solvothermal process, but the authors did not explicitly explain the difference between these two methods. In fact, these two methods are a little bit different. Authors should explore more. Is the solvothermal an eco-friendly method?
Author’s reply: Thank you for your careful view. We completely agree with you and the text is modified to clarify the differences between these two methods. According to the solvothermal technique, a lot of published papers focused on the incorporation of natural sources including plants are used in the green synthesis of water-soluble fluorescent carbon dots through hydrothermal/solvothermal treatment in a single step. This part was highlighted with red color in section 3.1 with confirmable references.
The authors mentioned that only three methods as eco-friendly processes and these methods are bottom-up methods. Authors should discuss how can these methods are eco-friendly compared to the top-down methods such as laser ablation, electrochemical reduction, etc.
Author’s reply: Thank you very much for your suggestion. Comparing between bottom-up and top-down techniques were inserted in the revised manuscript and marked with red color on page 2.
Authors are suggested to make a table for green starting materials used in the synthetic process. There are many new papers that have been published in 2019-2020 where carbon dots have been prepared using new natural and green starting materials by hydrothermal, pyrolysis, and microwave methods and add their references in the table.
Author’s reply: Thank you for your suggestion. An updated table has been inserted into the revised manuscript with recent references.
I found a word mistake inside Figure 3, correct it if there is a mistake. Please check Figure 8, I cannot see the full figure in the pdf file.
Author’s reply: Thank you very much for your concern. The exact word was revised, and Figure 8 was resized to be clear in pdf format.
In Figure 9, the authors discussed Ag-CQDs/SiO2/Si substrate. What is the purpose to discuss this material? It is better if the authors focus on carbon dots only not on their composites.
Author’s reply: Thank you for your comment. In Figure 9, we think that it follows the target of the review focusing on green synthesis (pyrolysis) and is confirmed by different examples.
The authors added several biomedical and biotechnological applications including cancer therapy and drug delivery, imaging and bioimaging, anti-microbial activity, and sensors. What is different between imaging and bioimaging. I cannot find it clearly in the text.
Author’s reply: Live cell bioimaging is becoming an increasingly popular tool for elucidation of biological mechanisms and is instrumental in unraveling the dynamics and functions of many cellular processes. Bioimaging is a method for imaging and direct visualization of biological processes in real-time which is often used to gain information on the 3D structure of the observed specimen from the outside, i.e., without physical interference.
Here, the addition of “sensors” part is not correlated to the biomedical applications. Authors should focus only on biomedical applications. It is recommended to remove “sensors” part. If the authors want to add “biosensors” part, then the title should change from biomedical applications to bio-applications.
Author’s reply: Thank you for your brilliant suggestion, the section of a biosensor is added to the text and the title changed accordingly.
Please make sure that the carbon dots applied in biomedical applications were synthesized using eco-friendly process. Authors should not randomly collect the references without the relation with eco-friendly synthetic process of carbon dots.
Author’s reply: Thank you very much for your valuable comment. We agree with you and the text is revised and focused on green synthesis.
Overall, the authors are recommended to comprehensively check the grammar mistakes, words, and chemical formula. Besides, the synthesis process of carbon dots should be discussed better and more compared to other parts, due to the manuscript submitted to the “processes” journal. If the authors focus on the writing of “process” part then we can understand the benefit of this article compared to various similar review articles.
Author’s reply: Thank you so much for your time reviewing our work and the manuscript is modified following your valuable advice.

Reviewer 3 Report
The authors report “New trends carbon quantum dot: ecofriendly process, properties, and biomedical applications”. Although the work presented is interesting to materials research community, this manuscript should be improved prior to its potential publication. Please find the minor concerns below.
- Colloidal synthesis is one of the main synthetic methodology to produce quantum dots. There are handful reports of CQDs using colloidal synthesis. The authors should consider this.
- I do not see a section for CQDs and GQDs physical characterization such as XRD, TEM, HRTEM, and other related techniques should be considered.
- I believe CQDs and GQDs are thermodynamically less stable when compared to the rest of the QDs. It would be interesting to see a section on this or the authors comment on stability somewhere.
Author Response
Reviewer 3"
Colloidal synthesis is one of the main synthetic methodology to produce quantum dots. There are handful reports of CQDs using colloidal synthesis. The authors should consider this.
Author’s reply: Thank you for your kind suggestion. In this review the main goal is a green synthesis for CQDs and GQDs which are based on carbon sources, so we have to correlate between CQDs and GQDs and we put the most appropriate techniques used in literature for the green approach to be easily comprehensible for the reader.
I do not see a section for CQDs and GQDs physical characterization such as XRD, TEM, HRTEM, and other related techniques should be considered.
Author’s reply: Thank you for your careful view. We used more than 10 Figures to fit the journal standard, so we focused on general procedures, properties, and applications to prevent paper crowded.
I believe CQDs and GQDs are thermodynamically less stable when compared to the rest of the QDs. It would be interesting to see a section on this or the authors comment on stability somewhere.
Author’s reply: Thank you for your question. The kind of crystalline carbon nanoparticle structure imagined by some in the literature is likely imagination only because there are no reasons in terms of thermodynamic driving forces to predict the formation of such a pretty but strangely ordered particle structure in the rather chaotic environment associated with the production methods discussed above.

Round 2
Reviewer 1 Report
The submitted version of manuscript „New trends carbon quantum dot: ecofriendly process, properties, and biomedical applications”, now under the title “Recent developments in carbon quantum dots; properties, fabrication techniques, and bio-applications” has been revised and some comments and suggestions have been addressed. Table 1 has been added, in which authors listed selected papers concerning mainly different synthesis approaches and properties of obtained CQDs or GQDs. Inclusion of the applications in this table was rather confusing (wave-like lines?). Separate Tables should be prepared for imaging and sensing applications. Some issues have been corrected but there are another incorrect sentences or descriptions. For example Fig. 12 has been explained incorrectly – page 11 lines 304-309 (fluorescence of CQDs is not quenched by pDNA but by AuNPs; pDNA plays a role of a bridge joining both cationic NPs due to electrostatic interactions!). Other example, Figure 7 has not been cited in the manuscript. Such incorrect or unclear explanations of mechanisms depicted in some figures are common. In general, grammar should be highly revised. Wording and phrasing should be revised too.
Concluding, the manuscript was not improved satisfactory to be recommend for publication.
Author Response
Reviewer 1#
The submitted version of manuscript „New trends carbon quantum dot: ecofriendly process, properties, and biomedical applications”, now under the title “Recent developments in carbon quantum dots; properties, fabrication techniques, and bio-applications” has been revised and some comments and suggestions have been addressed.
Author’s reply: Thank you very much for your time revising our work
Table 1 has been added, in which authors listed selected papers concerning mainly different synthesis approaches and properties of obtained CQDs or GQDs. Inclusion of the applications in this table was rather confusing (wave-like lines?). Separate Tables should be prepared for imaging and sensing applications.
Author’s reply: Thank you very much for your comment. Regarding the wave-like lines means that the green source is not applied in this application and is closed by wave line. Anyway, the table is modified to be separated for applications.
Some issues have been corrected but there are another incorrect sentences or descriptions. For example Fig. 12 has been explained incorrectly
Author’s reply: Thank you very much for your careful view. The figure has been explained correctly in the text.
page 11 lines 304-309 (fluorescence of CQDs is not quenched by pDNA but by AuNPs; pDNA plays a role of a bridge joining both cationic NPs due to electrostatic interactions!).
Author’s reply: Thank you again, the sentence is corrected.
Other example, Figure 7 has not been cited in the manuscript. Such incorrect or unclear explanations of mechanisms depicted in some figures are common.
Author’s reply: Thank you for your observation, the figure cited in the text and the mechanism was explained.
In general, grammar should be highly revised. Wording and phrasing should be revised too.
Author’s reply: Thank you for your advice. The whole manuscript was revised, including wording, phrasing, and grammar.

Reviewer 2 Report
The authors have responded my questions and have revised the manuscript based on my comments. I recommend this manuscript can be accepted in the present form.
Author Response
Reviewer 2#
Comments and Suggestions for Authors
The authors have responded my questions and have revised the manuscript based on my comments. I recommend this manuscript can be accepted in the present form.
Author’s reply: Thank you very much for your acceptance.

Round 3
Reviewer 1 Report
Supporting information should be removed.
Tables 1 and 3 seem to be cutted (right side).
Author Response
Reviewer 1#
Supporting information should be removed.
Author’s reply: Thank you for your comment, but I would like to inform you that our manuscript dos not contain supporting information
Tables 1 and 3 seem to be cutted (right side)
Author’s reply: Thank you for your carful view, all the table have been revised
